# Neuroprotective Effect of Antiapoptotic URG7 Protein on Human Neuroblastoma Cell Line SH-SY5Y

**DOI:** 10.3390/ijms25010481

**Published:** 2023-12-29

**Authors:** Ilaria Nigro, Rocchina Miglionico, Monica Carmosino, Andrea Gerbino, Anna Masato, Michele Sandre, Luigi Bubacco, Angelo Antonini, Roberta Rinaldi, Faustino Bisaccia, Maria Francesca Armentano

**Affiliations:** 1Department of Science, University of Basilicata, Viale dell’Ateneo Lucano, 10, 85100 Potenza, Italy; ilaria.nigro@unibas.it (I.N.); rocchina.miglionico@virgilio.it (R.M.); monica.carmosino@unibas.it (M.C.); roberta.rinaldi@unibas.it (R.R.); faustino.bisaccia@unibas.it (F.B.); 2Department of Biosciences, Biotechnologies and Biopharmaceutics, University of Bari, Via Orabona, 4, 70125 Bari, Italy; andrea.gerbino@uniba.it; 3Department of Biology, University of Padova, Via U. Bassi, 58/B, 35121 Padova, Italy; a.masato@ucl.ac.uk (A.M.); luigi.bubacco@unipd.it (L.B.); 4UK Dementia Research Institute at UCL, University College London, Tottenham Ct Rd, London W1T 7NF, UK; 5Department of Neurosciences, University of Padova, Via Belzoni, 160, 35121 Padova, Italy; michele.sandre.1@phd.unipd.it (M.S.); angelo.antonini@unipd.it (A.A.); 6Centro Studi per la Neurodegenerazione (CESNE), University of Padova, 35121 Padova, Italy

**Keywords:** endoplasmic reticulum (ER) stress, unfolded protein response (UPR), protein misfolding, PI3K/AKT pathway, autophagy, tunicamycin, apoptosis, neurodegeneration

## Abstract

Up-regulated Gene clone 7 (URG7) is a protein localized in the endoplasmic reticulum (ER) and overexpressed in liver cells upon hepatitis B virus (HBV) infection. Its activity has been related to the attenuation of ER stress resulting from HBV infection, promoting protein folding and ubiquitination and reducing cell apoptosis overall. While the antiapoptotic activity of URG7 in HBV-infected cells may have negative implications, this effect could be exploited positively in the field of proteinopathies, such as neurodegenerative diseases. In this work, we aimed to verify the possible contribution of URG7 as a reliever of cellular proteostasis alterations in a neuronal in vitro system. Following tunicamycin-induced ER stress, URG7 was shown to modulate different markers of the unfolded protein response (UPR) in favor of cell survival, mitigating ER stress and activating autophagy. Furthermore, URG7 promoted ubiquitination, and determined a reduction in protein aggregation, calcium release from the ER and intracellular ROS content, confirming its pro-survival activity. Therefore, in light of the results reported in this work, we hypothesize that URG7 offers activity as an ER stress reliever in a neuronal in vitro model, and we paved the way for a new approach in the treatment of neurodegenerative diseases.

## 1. Introduction

The maintenance of cellular protein homeostasis, defined as proteostasis [1], through the fine regulation and control of protein synthesis, trafficking, folding, secretion and degradation is crucial for proper cell functioning. The ER is the cellular compartment physiologically responsible, among other functions, for the synthesis, folding and quality control of at least one-third of cellular proteins, both resident in the ER or other organelles and those destined for secretion (after crossing Golgi apparatus), to favor intercellular communications. The malfunction of the ER is a condition that is referred to as ER stress and results in an accumulation in the lumen of unfolded or misfolded proteins. Cells try to counteract ER stress through the activation of a signaling network known as the Unfolded Protein Response (UPR), which aims to restore optimal proteostasis by modifying the expression of specific proteins, such as chaperones, by regulating protein synthesis and by affecting the protein degradation system to decrease the ER load [2]. Conversely, in prolonged cellular stress or an insufficient adaptive response, genes and pathways that lead to cell death are activated. The ability of cells to preserve such protein homeostasis as much as possible is susceptible to decline during ageing. Hence, there are several age-associated pathological conditions collectively referred to as “proteinopathies” or Protein Misfolded Disorders (PMD) [3], all characterized by the accumulation of proteins that aggregate or misfold in an unusual way. These metastable proteins (soluble oligomers and fibrillar amyloid deposits) are typically linked with neurodegeneration in Alzheimer’s and Parkinson’s diseases, and many other pathologies [4].

Recent studies carried out by our group characterized the activity of a protein defined as Up-regulated Gene clone 7 (URG7), which is mainly of hepatic origin and resident in the ER with an N-Lumen/C-Cytosol orientation [5,6]. The URG7 gene (745 base pairs, located on chromosome 16 in position “p13.1”) codes for a polypeptide of 99 amino acids, with the first 74 residues of the sequence overlapping with the N-terminal portion of the MRP6 protein, also known as ABCC6 (it is considered as its isoform 2), a protein belonging to the ABC (ATP-binding cassette) superfamily; the remaining 25 amino acids are peculiar to the URG7 protein [5]. The expression level of URG7 increases significantly during hepatitis B virus (HBV) infection, by the viral protein HBXAg (hepatitis B virus X antigen) transactivating activity [7,8], and its proven antiapoptotic activity has been related to the mitigation of ER stress, with a reduction in protein misfolding, favoring overall cell survival [9]. If the antiapoptotic effect of URG7 up-regulation in liver cells infected by hepatitis B virus could be considered deleterious, in neuronal cells the same effect could be protective with respect to neurodegeneration [10]. In order to verify this hypothesis, in the present study we evaluated URG7 activity in neuroblastoma-derived cells subjected to ER stress, chemically induced by the antibiotic tunicamycin, to increase the intracellular non-folded or misfolded protein content, a typical phenotype of neurodegenerative diseases. Our results confirmed the ability of URG7 to mitigate ER stress and to promote folding and cellular ubiquitination to eliminate the excess of incorrectly folded proteins, suggesting the need for further investigations on the possibility of a new therapeutic approach for neurodegenerative diseases.

## 2. Results

### 2.1. URG7 Expression in Mouse Brain Tissues and Primary Cells

We first sought to investigate whether the URG7 protein is expressed in the brain. According to the Barres Brain RNA-Seq platform (brainrnaseq.org, accessed on 23 May 2022), which provides RNA-seq data on sorted human and mouse cortical cells, ABCC6 presents very low expression levels in human neurons and astrocytes as compared to other neuronal proteins such as α-synuclein (about a 75-fold change in neurons) or ER markers such as calnexin (about a 140-fold change in neurons). Similarly, little expression of Abcc6 was reported in mouse brain cells. Nevertheless, little is known about URG7 expression in the brain. Hence, we decided to carry out preliminary assays to verify the URG7 expression in mouse brain tissues through Western blotting assays using an anti-URG7 antibody. We analyzed separately midbrain, striatal and cortical regions dissected from wild-type mice of 1–6–12 months of age (Figure 1A). Although little expression was detected in the midbrain and striatum (here identified by the dopaminergic markers tyrosine hydroxylase, TH, and dopamine- and cAMP-regulated phosphoprotein 32 kDa, DARPP-32), it appeared that URG7 showed an age-dependent increase in the cortex region. Based on this preliminary indication, we analyzed the differential expression of URG7 in primary mouse cortical cell cultures (Figure 1B). Notably, we detected URG7 by Western blotting in primary neurons, primary microglia and primary astrocytes, here identified by β3-Tubulin, Ionized calcium-Binding Adaptor molecule 1 (IBA1) and Glial Fibrillary Acidic Protein (GFAP), respectively. Overall, these data indicate a mild expression of URG7 in mouse-brain-derived tissues, possibly suggesting a role of URG7 in protein quality control in neuronal and glial cells.

### 2.2. Stable Clones’ Generation and Intracellular Localization of URG7 in SH-SY5Y Cells

To investigate the possible functional role of URG7 in the ER stress response in neuronal cells, an N-terminal-flagged URG7 construct was stably infected in the SH-SY5Y cell line. After selection with the proper selective agent (puromycin), the infection efficiency of the viral construct was evaluated by immunofluorescence experiments. As shown in Figure 2A, about 90% of cells resulted in being infected with pLV-URG7 and overexpressing the URG7 protein. The evaluation of the intracellular localization of URG7 was performed by immunofluorescence confocal microscopy. As shown in Figure 2B,C, this protein is localized around the nucleus (green signal), perfectly merging with the endoplasmic reticulum marker (calnexin, red signal), as evidenced by the yellow color in the merge. This result demonstrates that URG7 has an intracellular localization in the endoplasmic reticulum of SH-SY5Y cells.

In parallel with the immunofluorescence experiments, Western blotting assays were carried out to estimate the URG7 protein overexpression levels of the stably infected cell culture. Notably, the immunoblot performed by using an anti-URG7 antibody (Figure 3A, left panel) shows two URG7 protein bands that migrate between ~12 and ~17 kDa markers: the migration of the lower band fits well with the predicted mw of the URG7 protein (12 kDa, considered as endogenous protein, also responsible for the weak green signal in Figure 2A, pLV cell culture), while the upper band corresponds to the exogenous, Flag-tagged, overexpressed form of URG7, as confirmed by Western blotting analysis using an anti-Flag antibody.

### 2.3. Mitigation of ER Stress in URG7-SH-SY5Y Stable Clones

First, the level of the expression of the URG7 protein in SH-SY5Y stable cell lines was evaluated (infected with the pLV empty vector or with the pLV-URG7 vector), treated with the antibiotic tunicamycin (3 μg/mL) for 15 h to induce ER stress. Notably, tunicamycin treatment did not significantly influence the expression level of both the Flag-tagged URG7 protein and the endogenous one.

Subsequently, to evaluate the role of URG7 in the signaling pathway known as the unfolded protein response (UPR), the expression of different UPR markers was monitored by Western blotting assay, using protein lysates from SH-SY5Y stable clones exposed to tunicamycin.

As shown in Figure 3, the expression level of the main ER stress marker, i.e., the chaperone GRP78/BiP, was significantly altered, suggesting a strong UPR activation. Data analysis suggested the significant activation of all three branches of the UPR: the increased protein expression of sXBP1 and ATF4 indicates, respectively, the IRE1α and PERK branches’ activation, and the increase in the cleaved form of ATF6 indicates the activation of the same branch.

The expression of the homologous protein of the pro-apoptotic transcription factor C/EBP (CHOP) was also evaluated, and as shown in Figure 3, under stressed conditions and in the presence of URG7, its expression level resulted in being lower than in the control cells.

The authors highlight that URG7 expression alone did not trigger the UPR activation (as shown in Figure 3B).

These results clearly demonstrate that URG7 is able to modulate the protein expression pattern of UPR markers under ER stress conditions. The indication of a lower expression of the pro-apoptotic transcription factor CHOP allowed us to hypothesize that URG7 may also have a pro-survival role in this new cell model and motivated us to further investigate a putative role of URG7 in cellular apoptosis.

### 2.4. URG7 Favors Cell Survival and Inhibits Apoptosis in ER Stressed Cells

The involvement of URG7 in counteracting the apoptotic mechanisms triggered by the treatment of cell cultures with tunicamycin [11,12] was investigated. The expression level of the protein poly(ADP-ribose) polymerase (PARP), which is one of the main cleavage targets of the activated caspase-3, was evaluated under ER stress conditions. Western blotting analysis (Figure 4) showed a lower amount of the cleaved form (cPARP) in pLV-URG7 cells, suggesting that URG7 expression counteracts apoptosis activation. As shown in Figure 4, the expression of the antiapoptotic protein BCL2 was also significantly increased, while the expression of the pro-apoptotic protein BAX was down-regulated in pLV-URG7 cells.

An important intracellular signaling pathway able to promote cell survival is the PI3K/AKT pathway. Therefore, the level of AKT phosphorylation under ER stress conditions was evaluated, and as shown in Figure 4, a greater activation of the pathway was evident in pLV-URG7 cells compared to the control. β-catenin, a subunit of the cadherin protein complex, is the central hub of the Wnt\β-catenin pathway associated with cell survival: as shown in Figure 4, its expression was found to be significantly up-regulated, compared to the control, in the pLV-URG7 stable cell line.

In non-stressed conditions, most of the markers evaluated did not change significantly in the presence of URG7 (with the exception, in fact, of only cPARP and BCL2, which, also in this case, showed the same change as seen in the stressed conditions) and, therefore, are not reported in Figure 4B.

Therefore, these experimental evidences could actually suggest an antiapoptotic activity of the URG7 protein, through mechanisms that still need to be clarified.

### 2.5. URG7 Affects Protein Catabolism in ER Stressed Cells

Treatment of cell cultures with tunicamycin promotes ER stress and allows for the increase in unfolded/misfolded proteins within the cell. Their presence threatens cell survival if they are not adequately eliminated. In this study, the levels of misfolded proteins were quantified by labeling them with thioflavin T, a fluorescent dye widely used to visualize and quantify the presence of misfolded protein aggregates called amyloid, both in vitro and in vivo [13]. As shown in Figure 5A,B, the number of misfolded proteins in the tunicamycin-treated cells was significantly lower in stably expressing URG7 compared to their controls. Considering that the two main pathways regulating protein catabolism are the ubiquitin–proteasome system (UPS) and the autophagy–lysosomal system, the levels of cellular protein ubiquitination and the expression levels of two important autophagy markers were evaluated. As shown in Figure 5C,D, ubiquitination levels were significantly higher in the presence of URG7 after treatment with tunicamycin, indicating a higher UPS activation. Moreover, in this cell line, the expression levels of both autophagy markers, Beclin-1 and LC3-II, were also significantly higher (Figure 5E,F), suggesting that cellular proteins may also be catabolized through this mechanism.

### 2.6. URG7 Affects ER Calcium Depletion and Reduces ROS Production in SH-SY5Y Stressed Cells

Intracellular Ca^2+^ is stored mainly in the ER lumen. The use of agents such as tunicamycin, responsible for the induction of ER stress, has recently been shown to directly impact ER Ca^2+^ levels [14], also triggering mitochondrion-mediated apoptotic mechanisms [15]. Accordingly, we evaluated the potential effects of URG7 on the ER calcium Ca^2+^ levels. We then semiquantified the ER Ca^2+^ levels by expressing the FRET-based Ca^2+^ probe D1ER, in either URG7-expressing or control cells in resting and tunicamycin treatment conditions. As expected, tunicamycin-induced stress caused a significant decrease in ER Ca^2+^ in control cells. Interestingly, this drop in ER Ca^2+^ levels, upon tunicamycin treatment, was significantly reduced in the presence of URG7 (Figure 6A). This result further supports the hypothesis of an antiapoptotic role played by the URG7 protein. A decrease in Ca^2+^ within the ER triggers a process known as store-operated Ca^2+^ entry (SOCE), which causes Ca^2+^ influx through the plasma membrane. Two key components of SOCE are Stromal interacting molecule 1 (STIM1) and calcium release-activated calcium channel protein 1 (ORAI1). STIM1 is an ER Ca^2+^ sensor that in response to decreased ER Ca^2+^ concentrations, oligomerizes and translocates to ER-plasma membrane junctions where it associates with ORAI1, localized in the plasmatic membrane, forming highly Ca^2+^ selective pores [16]. Tunicamycin, by disrupting ER Ca^2+^ homeostasis, increased the expression of ORAI1 and STIM1 in control cells, but not in URG7 overexpressing cells, in agreement with the observed reduced ER Ca^2+^ depletion (Figure 6B,C). Under unstressed conditions, the evaluated markers did not change significantly in the presence of URG7 and, therefore, are not reported in Figure 6C.

It is well known that calcium depletion from the ER may be the basis for its accumulation in the mitochondria, which in turn causes the generation and release of reactive oxygen species, i.e., ROS [17]. Furthermore, ROS levels can increase in the ER itself under conditions of ER stress. Indeed, the overall intracellular ROS levels were measured under ER stressed conditions and the results are reported in Figure 6D: in the presence of URG7, the levels of ROS were found to be significantly lower than in the control cells.

## 3. Discussion

Many neurodegenerative diseases, such as Parkinson’s, Alzheimer’s, ALS, etc., share common hallmarks at the molecular level, including the accumulation and abnormal aggregation of misfolded proteins in the brain, also promoted by ER dysfunction. Neuronal cells are particularly sensitive to protein misfolding compared to non-neuronal cells, where cell duplications help counteract misfolding during ER stress by repeatedly diluting unfolded peptides. In contrast, non-dividing postmitotic neurons are completely dependent on the signaling pathway known as the unfolded protein response (UPR). If the misfolded proteins are not removed and normal cellular functions are not restored, the UPR triggers apoptotic pathways, leading to selective neuronal death [4,18].

We previously showed that URG7, an ER resident protein whose expression increased significantly in the case of HBV viral infection, acts as an ER stress reliever, promoting protein folding and ubiquitination, influencing UPR and reducing the overall level of cellular apoptosis in HepG2 stressed cells [9]. Furthermore, previous in vitro studies demonstrated the ability of URG7 to interfere with α-synuclein folding and aggregation [19]. Based on this rationale, we checked whether the URG7 protein could perform its peculiar functions in a different biological model, i.e., the neuronal one.

It is known that the highest URG7 expression levels are in the liver, while little is known on its specific expression in other human organs and tissues. Indeed, the first step of this study was to evaluate the endogenous expression levels of this protein both in immortalized neuronal cell lines, in primary neuronal cultures and in different brain regions (midbrain, striatum and cortex) in mice of different ages (1, 6 and 12 months). The acquired results highlighted the expression of URG7 predominantly in the cerebral cortex (Figure 1A) obtained from wild-type mouse brains, and a good level of expression both in primary and immortalized neuronal cell lines (Figure 1B).

Then, to better study the function of URG7 at the molecular level, we stably overexpressed URG7 in immortalized neuronal cell cultures (Figure 2A and Figure 3A), i.e., in the SH-SY5Y cell line, widely used in neurobiology as a valid in vitro cellular model [20,21]. Confocal microscopy analysis confirmed the intracellular localization of URG7 at the level of the endoplasmic reticulum (Figure 2B,C), as previously found in liver cells [5].

One of the main models used to mimic the biochemical and molecular processes altered during neurodegeneration consists of exposing neuronal cells to a compound with neurotoxic activity, such as tunicamycin, to mimic PD [21]. Therefore, in the present work, both SH-SY5Y cell cultures (the one able to overexpress URG7, pLV-URG7, and the control one, pLV) were treated with tunicamycin to cause ER stress, and subsequently the expression of several UPR markers was evaluated.

Our results demonstrated that URG7 overexpression per se does not generate ER stress compared to the control (Figure 3A). Conversely, treatment of both cell lines for 15 h with tunicamycin triggered the UPR, as demonstrated by the increase in protein expression of the measured markers. Nonetheless, the presence of URG7 was able to modulate the expression of these markers in a peculiar way compared to the control, increasing cell survival. Indeed, under tunicamycin treatment, the expression of the chaperonin GRP78/BiP (glucose-regulated protein 78/binding immunoglobulin protein) was 50% higher in the presence of URG7 than in control cells. This chaperonin serves several functions associated with cell survival, and in particular, it controls the folding and assembly of proteins within the ER and prevents the transport of incorrectly folded proteins/protein subunits [22,23]. Interestingly, among the etiopathogenetic factors of neurodegenerative diseases, the lack of maintenance of protein folding plays a crucial role, therefore suggesting the pharmacological regulation of the chaperones’ function as a therapeutic strategy [24]. We also demonstrated that all three arms of the UPR are activated by URG7. Moreover, both the Activating Transcription Factor 6 (ATF6) glycoprotein and the spliced X Box-Binding 1 (sXBP1) transcription factor (which derives from the branch of IRE1) are overexpressed: they allow the regulation of genes encoding chaperones and enzymes involved in the translocation, folding, maturation and secretion of misfolded proteins. Furthermore, they regulate the expression of proteins important in the functioning of the ERAD system, which is able to provide for the disposal of misfolded proteins [25,26]. The PERK (protein kinase-like endoplasmic reticulum kinase) branch seems to be the most activated since the expression of the transcription factor ATF4 (Activating Transcription Factor 4) increased more than ATF6 and sXBP1. ATF4 is responsible for the gene regulation of molecular chaperones and proteins involved in autophagy processes and is able to trigger both pro-survival and pro-death pathways in neurodegenerative diseases [27]. In our model, the overexpression of URG7 promotes cell survival rather than cell death, as suggested by the strong downregulation of the transcription factor CHOP (C/EPB homologous protein), responsible for the transcription of numerous genes involved in apoptotic mechanisms [28,29,30]. Moreover, as can be seen in Figure 4, in pLV-URG7 cells stressed with tunicamycin, the expression of BCL2 was significantly increased, while the expression of Bax protein was down-regulated, compared to the control. The expression pattern of these two proteins during ER stress is known in the literature to be an index of cellular protection [31,32].

One of the main mechanisms involved in neuronal survival is the PI3K/AKT pathway [33]: considering that URG7 is an activator of this pathway [9], AKT phosphorylation was evaluated in pLV-URG7 ER stressed cells. Figure 4 shows a higher level of AKT phosphorylation in the presence of URG7 compared to the control, which further confirms the pro-survival activity of URG7. It is also known that the activation, by phosphorylation, of AKT leads to the concomitant phosphorylation of the GSK3β protein, which is thus inactivated, contributing to the stabilization of β-catenin, a protein involved in cell adhesion, in the stabilization of the cytoskeleton and in the blockade of apoptosis. Pan et al. demonstrated that URG7, in liver cells, blocked apoptosis through the activation of β-catenin [34]: in neuronal cells (Figure 4), the activation of β-catenin is confirmed,further supporting the hypothesis that URG7 may exert an antiapoptotic activity in these cells.

Finally, the expression level of the protein poly(ADP-ribose) polymerase (PARP), which is one of the main targets of activated caspase 3 during apoptosis [35], was also evaluated under ER stress conditions. Protein expression analysis (Figure 4) showed, in pLV-URG7 stressed cells, a decreased amount of the cleaved form (cPARP) compared to control cells, again suggesting that URG7 expression counteracts the activation of apoptosis.

To further investigate the putative role of URG7 as an ER stress reliever, we verified whether the URG7 protein was also able to modify the amount of unfolded proteins within the cells performing a staining with ThT, a molecule able to bind protein aggregates, particularly amyloid fibrils, and then emitting fluorescence [13]. As demonstrated in Figure 5A,B, the total amount of unfolded/misfolded proteins upon tunicamycin treatment was lower in the presence of URG7 than in the control cells, suggesting a lowering of the overall protein load within the ER.

The whole pattern of protein ubiquitination in the two cell lines stressed with tunicamycin was evaluated and, as shown in Figure 5C,D, it is certainly higher in pLV-URG7 cells, confirming the ability of URG7 to increase protein degradation, preserving the cellular proteostasis [1].

The data obtained so far suggest the need to investigate the involvement of URG7 in autophagy, one of the mechanisms that allows neuronal cells to eliminate the excess of proteins under proteotoxic stress [36,37,38,39]. As can be seen in Figure 5E,F, under ER stress conditions, the up-regulation of both LC3-II and Beclin-1 proteins, both involved in the formation process of the autophagosome, was found in cells overexpressing URG7. Interestingly, the strong activation of ATF4 could explain the up-regulation of LC3-II and Beclin-1, as previously reported [40].

The imbalance of both cellular Ca^2+^ concentrations and redox status can reduce the protein folding ability of luminal chaperones within the ER, leading to the accumulation and aggregation of unfolded/misfolded proteins. A dysregulation of the balance Ca^2+^/ROS between the ER and the mitochondria is an established etiopathogenetic factor of various disorders, including protein misfolding pathologies [17,41,42,43]. Moreover, tunicamycin, by altering the homeostasis of the ER and triggering the UPR, induces an alteration in the homeostasis of Ca^2+^, which itself contributes to the UPR [14,44]. Therefore, the Ca^2+^ level within the ER under a stress condition, chemically induced by tunicamycin, was studied. As reported in Figure 6A, a significant decrease in Ca^2+^ in the lumen of the ER in pLV cells treated with tunicamycin was detected, as well as an increase in the expression of STIM1 and ORAI1 (Figure 6B,C), both key components of SOCE. These effects were much more limited in pLV-URG7 cells following tunicamycin treatment. Considering the important role of the intracellular second messenger played by Ca^2+^, mediating, among others, cell death mechanisms [15], this result further supports the hypothesis of an antiapoptotic role played by the URG7 protein.

The crosstalk between reactive oxygen species and Ca^2+^ is well known to play an essential role in many pathophysiological conditions, including neurodegenerative diseases [45]: for this reason, it was deemed appropriate to also evaluate the levels of ROS in the same conditions in which ER Ca^2+^ levels were quantified. Our results, reported in Figure 6D, highlight that the amount of ROS present in tunicamycin-stressed cells is lower in the pLV-URG7 cell culture than in the pLV control cells. This result once again suggests that the URG7 protein can play a role aimed at promoting cell survival as much as possible since a lower amount of ROS developing under ER stress certainly breaks the pro-apoptotic pathways.

Collectively, the data shown in this paper suggest that URG7 plays an important role in the complex signaling that governs and balances cell survival and cell death: it is able to mitigate ER stress, counteracting Ca^2+^ release from the ER and ROS production activating the ubiquitin–proteasome system (UPS) and autophagy, the two major intracellular protein degradation pathways. Our hypothesis is that URG7 exerts its activity by interacting with the molecular chaperone GRP78, its interactor (as shown in [9]) and main regulator of UPR, thus acting as an ER stress sensor, able to initiate different cellular and signaling pathways for restoring ER homeostasis and promoting overall cell survival. In recent years, in order to relieve ER stress, in addition to the use of pharmacological therapies (Salubrinal, Guanabenz, Sephin 1), preclinical studies have been conducted to evaluate the efficacy of gene therapies using, for example, the overexpression of GRP78 and sXBP1 [46]. In the first case, the use of specific molecules were used to inhibit or activate specific targets of the UPR (mainly the PERK pathway), resulting in an overall neuroprotective action. The use of recombinant viruses and/or protein and gene delivery systems [47] has become a promising emerging therapeutic option to avoid undesirable off-target pleiotropic effects in other organs, instead specifically targeting neurodegenerations. In this context, therefore, considering the promising results reported in this study and the further necessary evaluations that will certainly be carried out in the future, it is reasonable to also hypothesize the use of URG7 protein with this therapeutic target in neurodegenerative disorders such as Parkinson’s disease, where the pathological process affects numerous organs and tissues outside the brain [48].

## 4. Materials and Methods

### 4.1. Reagents and Antibodies

Dulbecco’s Modified Eagle’s Medium/Nutrient Mixture F-12 (1:1) (DMEM/F12-HEPES) medium was purchased from Corning (Corning, NY, USA). Dulbecco’s Phosphate-Buffered Saline solution (DPBS), penicillin–streptomycin solution and Fetal Bovine Serum (FBS) were obtained from EuroClone (Milan, Italy). Dimethyl sulfoxide (DMSO), Trypsin–EDTA Solution, Puromycin dihydrochloride, Propidium iodide, poly-L-lysine solution, Bovine Serum Albumin, 2′,7′-Dichlorofluorescin diacetate, Bradford Reagent and Tunicamycin (TN) were purchased from Sigma Aldrich-Merck (Saint Louis, MO, USA). Thioflavin T (ThT) was obtained from EMD Millipore (Burlington, Massachusetts, USA). Paraformaldehyde was purchased from Invitrogen (Waltham, MA, USA). RIPA buffer and Protease Inhibitor Cocktail were purchased from Cell Signaling Technology (CST, Danvers, MA, USA). Primary antibodies specific for poly-ADP-ribose polymerase (PARP, #9542), LC3A/B (I and II, #12741), CHOP (#2895), Ubiquitin (#3936), β-actin (#3700), anti-mouse IgG-HRP-linked (#7076), anti-rabbit IgG-HRP-linked (#7074), anti-rat IgG, HRP-linked (#7077), p-AKT (Ser473) (#4060), anti-AKT (#4691), β-Catenin (#8480) and Alexa 488-conjugated anti-rabbit IgG (#4412) were purchased from Cell Signaling Technology (CST, Danvers, MA, USA). Primary antibodies specific for BECLIN-1 (#849702), sXBP1 (#647501), ATF6 (#853101), ATF4 (#693902), BAX (#633602) and BCL2 (#633502) were obtained from Biolegend (San Diego, CA, USA). Primary antibodies specific for GRP78⁄ BiP (#ab21685) and anti-goat IgG Alexa Fluor 647 (# ab150131) were purchased from Abcam (Cambridge, UK). Primary antibodies specific for STIM1 (#66189-Ig) and ORAI1 (#66223-1-Ig) were obtained from Proteintech. Primary anti-calnexin antibody (#sc-23954) was obtained from Santa Cruz Biotechnology (Dallas, TX, USA). Primary antibodies specific for Vinculin (#AB6039), TH (#AB152) and DARPP32 (#AB10518) were purchased from Millipore-Merck (Saint Louis, MO, USA). Primary antibodies specific for β-actin (#A1978) and β3-Tubulin (#T8578) were purchased from Sigma Aldrich-Merck (Saint Louis, MO, USA). Primary antibodies specific for IBA1 (#019-19741) and GFAP (#Z0334) were purchased from Wako Chemicals (Neuss, Germany) and Dako (Santa Clara, CA, USA), respectively.

Lentiviral particles containing pLV-URG7 recombinant vector (pLV[Exp]-puro-CMV>FLAG/hABCC6 [NM_001079528.4], Vector ID: VB 190605-1150gzr, simply reported as pLV-URG7) and lentiviral control particles (pLV[Exp]-EGFP:T2A:Puro-EF1A>mCherry, reported as pLV) were designed by ourselves using the VectorBuilder online platform and were purchased from VectorBuilder Biotechnology Inc. (Chicago, IL, USA).

### 4.2. Animals, Brain Lysis and Primary Cultures

Housing and handling of wild-type C57BL/6J mice were performed in compliance with national guidelines. All animal procedures were approved by the Ethical Committee of the University of Padova and the Italian Ministry of Health (#200/2019-PR). Midbrain, striatal and cortical brain regions were collected at 1–6–12 months of age and mechanically lysed in 25 mM Tris-HCl pH 7.5, 150 mM NaCl, 1% NP-40, 1% sodium deoxycholate, 0.1% SDS, 2 mM EGTA, 20 mM sodium fluoride, 50 mM beta glycerophosphate, 50 mM sodium pyrophosphate, 20 mM sodium orthovanadate [49]. Primary mouse neuronal, microglia and astrocytic cultures were obtained from P0-1 wild-type C57BL/6J mice as previously described [50,51]. At 12 days in vitro (DIV12), primary cells were harvested and lysed in RIPA buffer for further analysis by Western blotting.

### 4.3. Cell Culture and Treatments

The immortalized neuroblastoma cell line SH-SY5Y was purchased from the Cell Factory-ICLC (Genova, Italy; cell line catalog code: HTL95013). The cells were maintained in DMEM/F-12-HEPES medium containing FBS 10%, penicillin 100 U/mL and streptomycin 100 μg/mL, at 37 °C and 5% CO_2_, and were passaged every 48 h. In order to induce ER stress, SHSY5Y cell line stably expressing URG7 and control cells were treated with 3 µg/mL of tunicamycin (TN) for 15 h [9].

### 4.4. URG7 Stable Cell Line Generation

The cell line SH-SY5Y was engineered to stably overexpress URG7 protein. SH-SY5Y cells were plated at a density of 2 × 10^5^ cells/well into a 12-well plate and after 24 h were transduced, using antibiotic-free growth medium for 48 h, with lentiviral particles containing pLV-URG7 recombinant and pLV control plasmids at 20 multiplicity of infection, according to manufacturing instructions. Two days later, positively pooled cell lines were seeded in a 10 cm dish and cultured in the presence of 2 µg/mL puromycin for 14 days. Resistant clones were subsequently screened for URG7 expression by immunofluorescence and Western blotting assays. The clone with the highest URG7 expression was expanded and used for all experiments.

### 4.5. Immunofluorescence and Confocal Microscopy

SH-SY5Y cells, stably transduced with pLV and pLV-URG7 vectors, were grown on poly-L-lysine-coated coverslips till 50% of confluence, fixed with 4% paraformaldehyde for 10 min at room temperature (RT), permeabilized with 0.2% Triton X-100 in PBS and blocked with 1% BSA in PBS (saturation buffer). Subsequently, cells were incubated with anti-URG7 antibody [52] diluted 1:200 in saturation buffer for 2 h at RT. After 3 washes in PBS, cells were incubated with Alexa 488-conjugated anti-rabbit IgG diluted 1:1000 in saturation buffer for 1 h at RT. Following three washes, nuclei were stained with 1.5 μM propidium iodide in PBS and images were captured using FLoid CellTM Imaging Station (Thermo Fisher Scientific). For subcellular colocalization analysis, cells were also incubated with anti-calnexin antibody diluted 1:400 in saturation buffer for 2 h at RT and subsequently with Alexa 647-conjugated anti-goat IgG diluted 1:1000 in saturation buffer for 1 h at RT. Confocal images were obtained with a laser scanning fluorescence microscope Leica TCS-SP2 (HCX PL APO, 63x/1.32–0.60 oil objective); 8-bit images were saved at 1024 × 256 and acquired using the Leica Confocal Software (LCS Lite 2.0 version).

### 4.6. Western Blotting Analysis

SH-SY5Y cells stably expressing URG7 and mock transduced cells were plated in 6-well plates at a seeding density of 5 × 10^5^ cells/well and treated with 3 μg/mL of tunicamycin for 15 h to induce ER stress. Cells were lysed in RIPA buffer, supplemented with Protease Inhibitor Cocktail and protein concentration was measured by Bradford assay [53]. In total, 40 µg of each sample was resolved on SDS-PAGE and transferred onto nitrocellulose membrane. The membranes were incubated for 1 h at room temperature in blocking buffer containing 5% *w*/*v* non-fat dry milk in TRIS-Buffered Saline pH 7.4 and 0.1% *v*/*v* Tween 20 (TBST), followed by incubation with specific primary antibodies diluted in blocking buffer at 4 °C overnight (anti-URG7, anti-Vinculin, anti-TH, anti-DARPP32, anti-β3-Tubulin, anti-Iba1, anti-GFAP, anti-BiP/GRP78, anti-CHOP, anti-sXBP1, anti-PARP-1/cPARP-1, anti-β-catenin, anti-ubiquitin, anti-Beclin-1, anti-LC3 A/B, anti-STIM1 and anti-β-actin 1:1000, anti-ATF4 1 μg/mL, anti-ATF6 5 μg/mL, anti-AKT 1:100, anti-pAKT 1:2000, anti-BAX 1 μg/mL, anti-BCL2 1 μg/mL and anti-ORAI1 1:5000). Each membrane was then incubated with the appropriate secondary antibody horseradish peroxidase (HRP) conjugated for 1 h at room temperature, and immunoreactive bands were detected by the Chemidoc XRS detection system (BioRad) with the ImageLab software (4.0 version), using enhanced chemiluminescence reagents (ECL Star Enhanced Chemiluminescent Substrate or LiteAblot TURBO Extra Sensitive Chemiluminescent Substrate (EuroClone)). Densitometric analysis of protein bands was performed using Image J software (1.50i version). Expression of β-actin or Vinculin was used as a loading control.

### 4.7. Determination of Intracellular Reactive Oxygen Species (ROS)

To measure the intracellular ROS level, 1 × 10^5^ SH-SY5Y cells stably expressing URG7 and mock transduced cells were seeded into 24-well plates, exposed to 3 μg/mL of tunicamycin for 15 h and finally incubated with 1 μM 2,7-dichlorodihydrofluorescein diacetate (DCFH-DA) for 30 min at 37 °C in a humidified 5% CO_2_ atmosphere. The cells were then harvested by trypsinization, resuspended in PBS and analyzed by FACSCanto II flow cytometer (BD Biosciences), using 488 nm excitation and 530 nm emission wavelengths.

### 4.8. Measurement of Intraluminal ER Ca^2+^

Steady state FRET experiments were carried out using MetaMorph software (7.0 version, Molecular Devices, MDS Analytical Technologies, Toronto, Canada) in cells transfected with cameleon (D1ER) probe, which contains calreticulin signaling sequence flanked by the ECFP, calcium-binding calmodulin domain (D1) and citrine to monitor the ER calcium stored [54]. ECFP and citrine were excited at 435 or 509 nm, respectively. FRET from ECFP to citrine was determined by excitation of ECFP and measurement of fluorescence emitted from citrine. For steady state FRET experiments, FRET values were calculated as NetFRET according to the formula: NetFRET signal = FRET signal − a × citrine signal − b × ECFP signal, where a and b are the ratios of the signal in the FRET channel to the signal in the citrine channel in the absence of a donor and to the signal in the ECFP channel in the absence of an acceptor, respectively.

### 4.9. Statistical Analysis

Data were presented as means ± Standard Error Measurement (SEM) of three independent experiments, each performed in triplicate. For statistical analysis, GraphPad Prism software was used (version 8.4.2, GraphPad Software, San Diego, CA, USA). Significant differences between means were tested by one-way analysis of variance (ANOVA) followed by Dunnett’s post hoc test or by unpaired, two-tailed *t*-test. (*p*-values < 0.05 were considered as statistically significant).

## 5. Conclusions

In summary, in this paper, the authors confirmed, in the context of a neuronal in vitro model, the pro-survival activities of the URG7 protein, also ascertaining the interference with the release of Ca^2+^ from the ER and with the production of ROS, and the activation of the ubiquitin–proteasome system (UPS) and autophagy, the two main pathways of intracellular protein degradation. Taking into account that the association between ER stress, UPR and neuropathology is well established, it is reasonable to hypothesize that a pharmacological modulation of UPR in the affected tissues, mediated in this case by a protein, could potentially contribute to the treatment and prevention of neurodegeneration. While aware of the inevitable limitations and shortcomings present, which we will certainly overcome and fill with further experimental investigations using models, even in vivo models, that will be increasingly similar to pathological ones, we believe that these new discoveries clarify, for the first time, the molecular mechanisms underlying the activity of this small protein, which can be usefully exploited in a new cellular context and, therefore, give it significant therapeutic prospects.

## Figures and Tables

**Figure 1 ijms-25-00481-f001:**
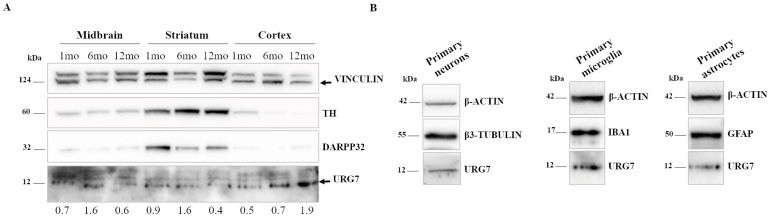
Expression of URG7 in mouse brain tissues and primary mouse cultures. (**A**) Western blotting analysis of URG7 in wild-type mouse midbrain, striatum and cortex at 1–6–12 months of age. Vinculin is used as loading control, while TH and DARPP32 are markers of the dopaminergic pathway in the midbrain and striatum. The relative level of URG7 expression, detected in different brain regions at different ages, was obtained by quantifying the intensity ratio of the URG7/Vinculin band. (**B**) Western blotting analysis of URG7 expression in primary cultures dissected from wild-type mouse cortical region. β3-Tubulin, IBA1 and GFAP were used as markers of neurons, microglia and astrocytes, respectively, and β-actin was used as the loading control.

**Figure 2 ijms-25-00481-f002:**
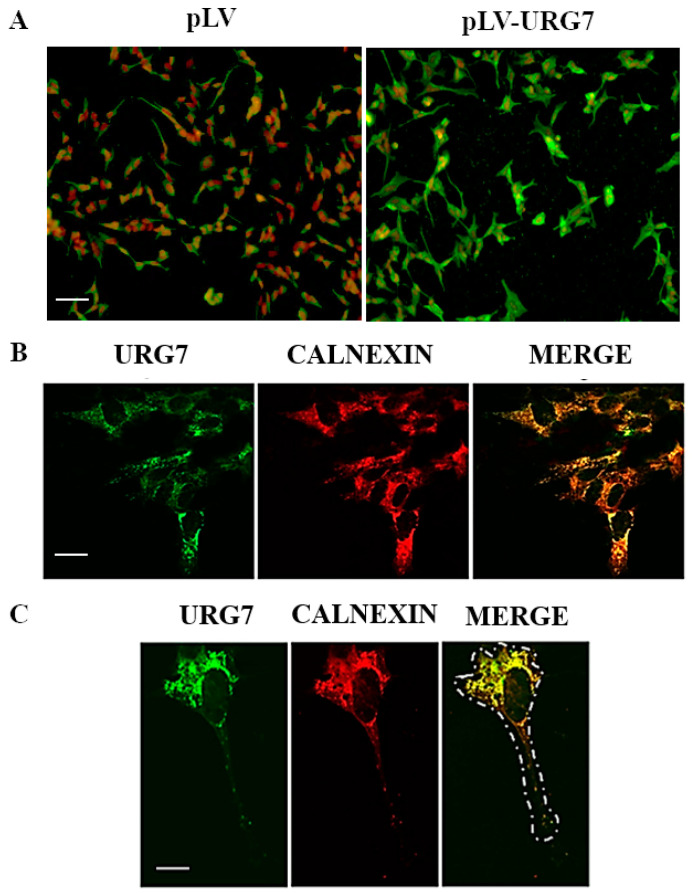
Characterization of URG7-SH-SY5Y stable clones. (**A**) Immunofluorescence detection of URG7 in SH-SY5Y cells stably infected with either pLV vector or pLV-URG7 vector. Cells were stained with anti-URG7 antibody (green signal) and nuclei were stained with propidium iodide (red signal); bar = 15 µm. (**B**) Laser confocal immunofluorescence analysis of URG7 (green) and calnexin (red) in URG7 SH-SY5Y clones. Colocalization of the two proteins is shown in yellow in the merge panel; bar = 10 µm. (**C**) Laser confocal immunofluorescence analysis at high magnification of URG7 (green) and calnexin (red) in URG7 SH-SY5Y clones; bar = 5 µm. The dashed line in merge panel represents the cell contour.

**Figure 3 ijms-25-00481-f003:**
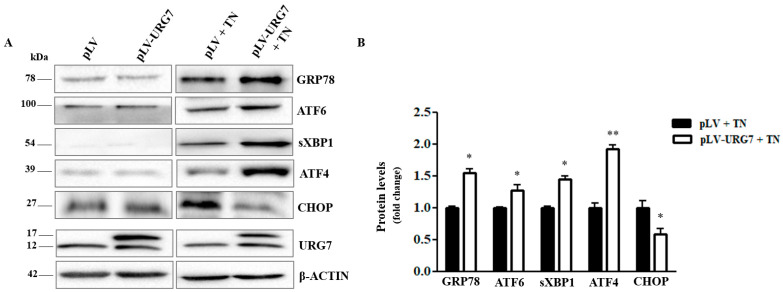
Effect of URG7 on ER stress markers’ expression levels. (**A**) Representative Western blotting and (**B**) densitometric analysis performed in SH-SY5Y cells stably infected with pLV vector and with recombinant pLV-URG7 vector treated with 3 μg/mL of tunicamycin (TN) for 15 h. β-actin was used as the loading control. All data, expressed as means ± Standard Error Measurement (SEM) of three independent experiments, are shown as fold change compared with the control value (mock). Statistical analysis was evaluated using GraphPad Prism 8.4.2 software (unpaired, two-tailed *t*-test, * *p* < 0.05, ** *p* < 0.01).

**Figure 4 ijms-25-00481-f004:**
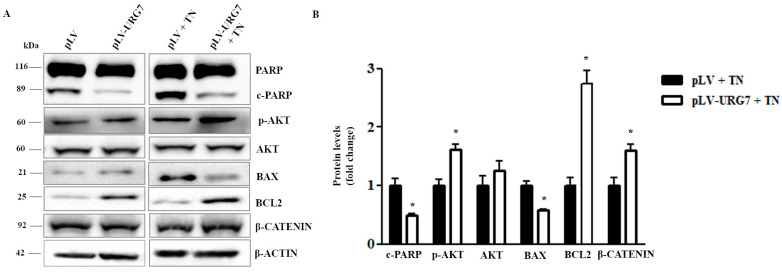
Antiapoptotic activity of URG7 in SH-SY5Y cells treated with tunicamycin. (**A**) Representative Western blotting and (**B**) densitometric analysis performed in SH-SY5Y cells, stably infected with pLV and with recombinant pLV-URG7 vectors, treated with 3 μg/mL of tunicamycin (TN) for 15 h. β-actin was used as the loading control. All data, expressed as means ± Standard Error Measurement (SEM) of three independent experiments, are shown as fold change compared with the control value (mock). Statistical analysis was evaluated using GraphPad Prism 8.4.2 software (unpaired, two-tailed *t*-test, * *p* < 0.05).

**Figure 5 ijms-25-00481-f005:**
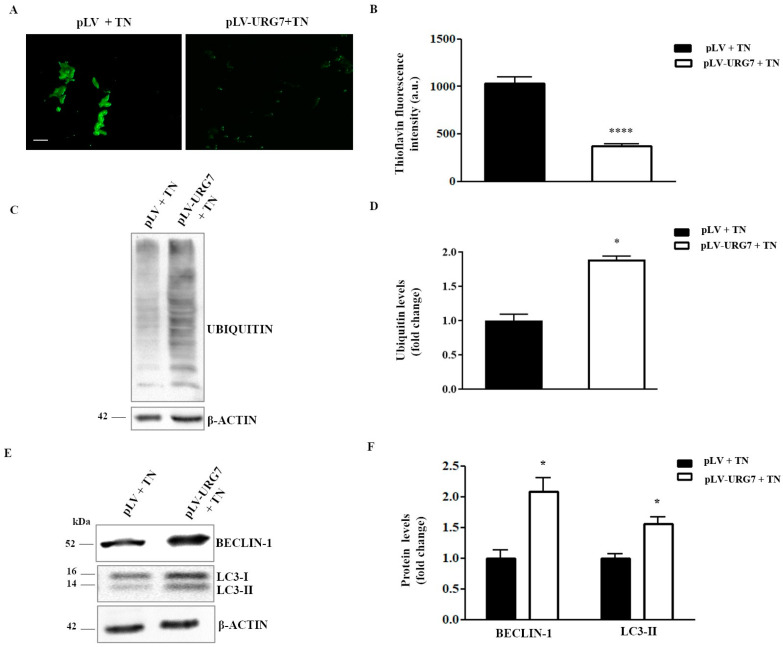
URG7 affects protein catabolism in stressed conditions. (**A**) Representative images and (**B**) fluorescence intensity analysis of stably infected SH-SY5Y cells treated with 3 μg/mL of tunicamycin (TN) for 15 h, stained within the last 3 h with 2.5 µM thioflavin T (green signal) and observed using FLoid CellTM Imaging Station (Thermo Fisher Scientific, Waltham, MA, USA); bar = 15 µm. (**C**) Representative Western blotting and (**D**) densitometric analysis of ubiquitinated proteins measured in stably infected SH-SY5Y cells after tunicamycin treatment. β-actin was used as the loading control. (**E**) Representative Western blotting and (**F**) densitometric analysis of autophagy markers performed in SH-SY5Y clones treated with 3 μg/mL of tunicamycin for 15 h. β-actin was used as the loading control. All data, expressed as means ± Standard Error Measurement (SEM) of three independent experiments, are shown as fold change compared with the control value (mock). Statistical analysis was evaluated using GraphPad Prism 8.4.2 software (unpaired, two-tailed *t*-test, * *p* < 0.05, **** *p* < 0.0001).

**Figure 6 ijms-25-00481-f006:**
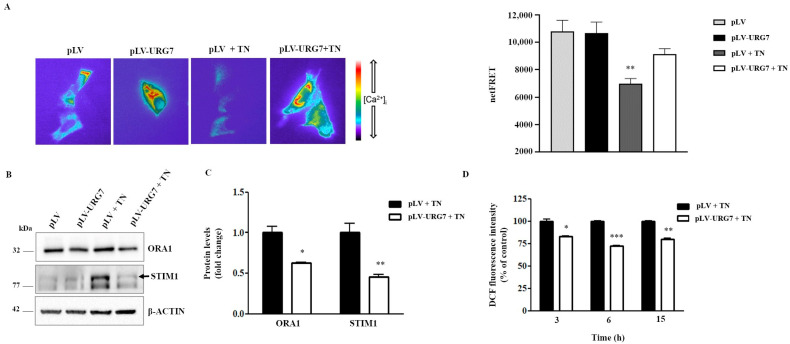
Evaluation of [Ca^2+^] ER with ER-targeted cameleon (D1ER) probe and evaluation of intracellular reactive oxygen species (ROS) level in stressed SH-SY5Y cells stably expressing URG7 protein. (**A**) The steady state D1ER FRET signal (ratio 530/470 nm) in SH-SY5Y cells expressing either the empty vector (pLV) or URG-7 vector (pLV-URG7) treated or not with tunicamycin (TN) is depicted in pseudocolor. The histogram compares changes in netFRET in control cells in both basal conditions (pLV; *n* = 31 cells) and after TN treatment (pLV + TN; *n* = 46 cells) and in URG-7-expressing cells in both basal conditions (pLV-URG7; *n* = 62 cells) or after TN treatment (pLV-URG7 + TN, *n* = 71 cells). Data are expressed as means ± Standard Error Measurement (SEM). Statistical analysis was performed on three independent experiments and significance calculated by ANOVA. ***p* < 0.01 pLV + TN vs. pLV. (**B**) Representative Western blotting and (**C**) densitometric analysis performed in SH-SY5Y clones treated with 3 μg/mL of TN for 15 h. β-actin was used as the loading control. Results, expressed as means ± Standard Error Measurement (SEM) of three independent experiments, are shown as fold change compared with the control value (mock). Statistical significance was evaluated using GraphPad Prism 8.4.2 software (unpaired, two-tailed *t*-test, * *p*< 0.05, ** *p* < 0.01,). (**D**) SH-SY5Y cells stably transfected with pLV and pLV-URG7 were treated with TN for 3, 6 and 15 h and the levels of intracellular ROS were detected using DCFH-DA fluorescent probe by FACScan flow cytometry. All data, expressed as means ± Standard Error Measurement (SEM) of three independent experiments, are shown as % of the control value (mock) and statistical significance was evaluated using GraphPad Prism 8.4.2 software (unpaired, two-tailed *t*-test, * *p* < 0.05, ** *p* < 0.01, *** *p* < 0.001).

## Data Availability

Data are contained within the article.

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
