# Peer review of "Neuroprotective Effect of Antiapoptotic URG7 Protein on Human Neuroblastoma Cell Line SH-SY5Y"

_ijms, 2023, doi:10.3390/ijms25010481_

Round 1

Reviewer 1 Report

Comments and Suggestions for Authors

The paper is written clearly and easily understood. The results and methodology are presented systematically. The idea seems interesting. I suggest the following corrections to improve it:

- Abstract: lines 27 and 33: in vitro (italic); provide statistical p values.

- Keywords: ER, UPR – provide full names. Generally, keywords and headings should contain full names (not only abbreviations)

- Introduction, lines 39 and 40 - protein synthesis, cell functioning.

- Introduction, the last paragraph, “new therapeutic approach for neurodegenerative diseases” – the authors should provide more details on this possibility and the importance of this paper compared to the literature.

- Discussion: 1) Provide more data on the current therapies for neurodegenerative diseases and compare with this study. 2) Then, please discuss if this (or similar) idea for the treatment of neurodegenerative diseases was already reported, or this is the first time. 3) In addition, please provide more details on explaining the proposed therapy.

- Statistical analysis: “means ± Standard Error (SE)” - “means ± Standard Error Measurement” (SEM)”.  

Reviewer 2 Report

Comments and Suggestions for Authors

Reviewer 3 Report

Comments and Suggestions for Authors

The reviewed work is devoted to the development of neuroprotective therapy for neurodegenerative diseases. The authors investigate the neuroprotective effect of up-regulated Gene clone 7 (URG7) on the SH-SY5Y cell line with tunicamycin-induced ER stress.

-The main remark is that the authors, while studying a number of biochemical parameters, changes in which should ultimately lead to greater cell survival, do not show the neuroprotective effect of URG7 itself. The work does not provide data on cell death either under ER stress or when using URG7.

-Page 5, lines 194 – 195. “Under unstressed conditions, the evaluated markers did not change significantly in the presence of URG7 and, therefore, were not reported in the graph 4B.” However, this is not entirely true when it comes to the expression of cPARP-1 and BCL2, as shown in Figure 4A.

- It has now been shown that Thioflavin T is not a highly specific dye for detecting amyloid proteins, so Congo red is usually used in addition (or instead).

- Oxidative stress analysis was carried out at several time points. “Control” is the ROS level at a given time point, taken as 100%. This presentation of the result does not reflect the dynamics of the development of the process, both under stress and when using URG7.

Minor remarks

- The “Conclusions” section would look more logical immediately after the “Discussion” section.

- The names of the proteins in Figures 4 and 5 and in the text should be consistent.

- Figure 1. After staining with antibodies to URG7, a green signal is observed in both the “pLV vector” and “pLV-URG7 vector” variants, which indicates the expression of URG7 in this cell type. This should be indicated in the text.

Comments on the Quality of English Language

Minor editing of English language

Round 2

Reviewer 3 Report

Comments and Suggestions for Authors

No questions or comments.

Comments on the Quality of English Language

 Minor editing of English language